# Could APTIMA mRNA Assay Contribute to Predicting Cervical Bacterial Sexually Transmitted Co-Infections? A Colposcopy Population Study [note 1]

**DOI:** 10.3390/ijms252313146

**Published:** 2024-12-06

**Authors:** George Valasoulis, Abraham Pouliakis, Ioulia Magaliou, Dimitrios Papoutsis, Nikoletta Daponte, Chrysoula Margioula-Siarkou, Georgios Androutsopoulos, Alexandros Daponte, Georgios Michail

**Affiliations:** 1Department of Obstetrics & Gynaecology, Faculty of Medicine, School of Health Sciences, University of Thessaly, 41500 Larisa, Greece; 2Department of Midwifery, School of Health Sciences, University of Western Macedonia, 50100 Kozani, Greece; 3Second Department of Pathology, National and Kapodistrian University of Athens, Attikon University Hospital, 12462 Haidari, Greece; 4Gynaecologic Oncology Unit, Second Department of Obstetrics and Gynaecology, Aristotle University of Thessaloniki, 54124 Thessaloniki, Greece; 5Department of Obstetrics & Gynaecology, Faculty of Medicine, School of Health Sciences, University of Patras, 26504 Patras, Greecegmichail@upatras.gr (G.M.)

**Keywords:** HPV DNA, HPV mRNA, mRNA E6/E7, APTIMA, sexually transmitted infections, bacterial STIs, colposcopy patients, cervicitis, inflammatory changes, cervical screening

## Abstract

In addition to chronic hrHPV anogenital infection, continuing inflammatory cervical changes are intrinsic in the development of precancerous lesions. In younger women, much of this inflammatory background parallels the progressive maturation of squamous metaplasia, often rendering treatment interventions redundant; however, patients with persistent cervical precancer, as well as those harboring invasive bacterial pathogens, might benefit from controlling the active inflammatory process by shortening the HPV natural cycle and avoiding subsequent cervical surgery. In a colposcopy population of 336 predominantly young asymptomatic individuals, we explored the impact of molecularly detected bacterial STIs on HPV DNA and APTIMA positivity rates using validated assays. In the multivariable analysis, several largely anticipated epidemiological factors were related to STI positivity. In this cohort, the HPV DNA test illustrated better performance for the prediction of STI positivity than the corresponding APTIMA test (sensitivity 52.94% vs. 33.82%), while inversely, the APTIMA test was more indicative of bacterial STI negativity than the HPV DNA test (specificity 77% vs. 60%). In addition, no significant differences between these two molecular assays were documented in terms of PPV, NPV, and overall accuracy. Despite the high *Ureaplasma urealyticum* and low *Chlamydia trachomatis* prevalence recorded in this study’s population, which is among the first assessing the co-variation of bacterial STI expression with established HPV biomarkers, the APTIMA assay did not predict concurrent bacterial STIs superiorly compared with an established HPV DNA assay.

## 1. Introduction

Together with persistent hrHPV infection, virulent cervical inflammatory changes (either in the context of immune-mediated chronic cervicitis or resulting from bacterial pathogen infections) are intrinsic in the development of cervical precancerous lesions, especially in younger individuals. Cervicitis has been associated with the depletion of cervical columnar cells, a typical feature of the maturational process; thus, STIs could represent squamous metaplasia promoters [1,2]. While most hrHPV-induced LSIL will spontaneously regress, recent research has mostly focused on alterations of vaginal microbiome and related microecology as triggers of hrHPV-related carcinogenesis; however, the influence of cervical bacterial STI co-infections on cervical precancer natural history is often overlooked [2,3].

Evolution in cervical prevention strategies has mandated a global shift towards HPV-based approaches, mainly affecting older individuals undergoing screening. This change is attributable to the superior performance and cost-effectiveness of validated molecular methods, the compatibility with self-sampling approaches, as well as the gradual incorporation of vaccinated cohorts in cervical screening [4]. Despite HPV DNA assays being mostly utilized in primary cervical screening, strategies implementing validated mRNA HPV platforms also perform favorably [5,6].

Based on previous clinical observations, we hypothesized that the HPV mRNA results might fluctuate with bacterial STI positivity and potentially identify individuals with more severe cervical lesions. The aim of this study, conducted in a previously described cohort, is to assess subtle differences in hrHPV DNA and APTIMA HPV positivity rates in individuals testing positive for specific bacterial STIs such as *Chlamydia trachomatis* (Ct), *Mycoplasma genitalium* (Mg), *Mycoplasma hominis* (Mh) and *Ureaplasma* spp. With the widespread use of multiplex PCR platforms currently being utilized in routine cervical screening globally, our ultimate ambition would be to contribute to the refinement of management algorithms for hrHPV-positive individuals with prevalent bacterial STI, contributing to a more tailored patient approach.

## 2. Results

### 2.1. Demographic Data

Demographic characteristics and medical history of the 336 women who participated in this study are presented in Table 1. The median age was 28 years (Q1–Q3: 24–34 years, minimum 18 years, maximum 48). About one in three (36.9%) participants were smokers, and a similar percentage (34.5%) were vaccinated, predominantly (81%) with the quadrivalent Gardasil vaccine. The average number of lifetime sexual partners was 6; about one-third (33.9%) had ≤3 lifetime partners and a similar percentage (36.3%) >5. The average percentage of condom use was 29.8%; 150 women were never users (44.6%), and 50 (14.9%) were consistent users (80% or higher rate of condom utilization, detailed condom use percentages as reported are illustrated in Table 1).

Clinical data and cervical infection status characteristics of this study are presented in Table 2. More than half of the individuals (180/336, 53.6%) presented aberrant cytology results, while 210 women (62.5%) had abnormal colposcopic findings. HPV DNA was positive in 152 (45.2%) of the participating women; of these, 60 (39.5% of the total HPV-positive population) had co-infection with multiple HPV genotypes. A total of 92 women (27.4%) were found APTIMA positive; notably, no further individual mRNA genotyping was performed. Finally, a history of genital warts (clinically evident or previously treated) was reported in 18 women (5.4%). The documented distribution of individual HPV genotypes reflects previously reported geographical data [7,8]; more details for HPV DNA genotyping have been presented in our previous works [9,10,11,12].

Focusing on HPV DNA genotype analysis, we estimated that from the HPV DNA-positive women, 36.8% would have been fully covered by the nonavalent vaccine if duly vaccinated, 39.5% would not have been covered, and 23.7% would benefit from partial coverage.

One hundred thirty-six (136) women (40.5%) were positive for bacterial STIs (Ct, Mg, Mh, or *Ureaplasma* spp.). The most prominent infection was *Ureaplasma* spp., with 122 women (36.3% of the participating women) harboring this pathogen. A significant 89.7% of the bacterial STI-infected women were *Ureaplasma* spp. positive, and in 20 individuals, multiple bacterial infections were identified (see Table 2).

### 2.2. Factors Affecting STIs Positivity

#### 2.2.1. Univariate Analysis

We subsequently investigated the relation of a positive vs. negative bacterial STI test outcome with other recorded patient characteristics. Odds Ratios (ORs) are presented whenever possible. The results are analyzed in Table 3, where the second column reports the contingency table between an STI negative or positive result and the values of each studied parameter. Individual LBC outcomes were linked to STI outcomes; specifically, 50% of the STI-negative women also exhibited normal cytology, while this percentage was 41.2% for women harboring an STI (*p* = 0.0152). Abnormal colposcopic findings were related to a positive STI test (*p* = 0.0019) (See Table 3), particularly 96 of the 136 individuals (i.e., 70.6%) with positive STI results presented with abnormal colposcopy; the remaining 40 (29.4%) had normal colposcopy. Similarly, 57% of the women not harboring an STI presented with abnormal cytology (OR: 1.8, 95%CI: 1.1–2.6). STI positivity was related to HPV DNA positivity (OR: 1.7, 95% CI: 1.1–2.6), as well as with APTIMA positivity with similar ORs: 1.7 (95% CI: 1.1–2.8); however, STI positivity rates in the analysis were not related to genotypes that could be potentially covered by the nonavalent vaccine (*p* = 0.1318).

The number of sexual partners also illustrated statistical significance for STI positivity for almost all thresholds; women positive for STI had more lifetime sexual partners than the specified threshold. Interestingly, the lowest *p*-value was for a more than one (≥1) sexual partner cut-off (compared with the other cut-offs, see Table 3), which was also linked with the highest odds for STI positivity (OR: 6.3, 95% CI: 2.2–18.2) and indicative that women with a history of more than one sexual partner, were more likely to be tested positive for STIs. Similar outcomes were observed for a recent (i.e., during the previous year) partner change (OR: 2.3, 95% CI: 1.3–4), suggesting that women with STIs had more than double the odds of undergoing a partner change during the last year before inclusion in this study.

Interestingly, smoking status, history of genital warts, and condom use did not reach statistical significance in our study (*p* > 0.05 for all parameters).

Remarkably, timely HPV vaccination was linked with diminished risk for STI detection since only 23.9% of the STI-positive women had been previously vaccinated, whereas, in the STI-negative women, this percentage almost doubled (42%); therefore, HPV-vaccinated women indeed seemed to be at less risk for other concurrent bacterial sexually transmitted infections.

#### 2.2.2. Multivariable Analysis

Women’s characteristics identified as important in the univariate analysis with a *p*-value < 0.2 were entered into a multivariable analysis model to control for possible confounders. Because of the limited number of cases with HSIL histology (<10 patients), a detailed LBC and colposcopy outcomes correlation assessment was not considered since cases with NILM cytology and normal colposcopy were disproportionally represented in this study’s sample. Figure 1 illustrates a diagram of the Odds Ratios (ORs) for those parameters, illustrating statistical significance using as a reference point a positive STI test.

Interestingly, in the multivariable analysis (see Table 4), the number of term pregnancies emerged as an independent predictor of STI positivity (OR: 2.16, 95% CI: 1.25–3.73, *p* = 0.00760), indicative that with higher parity, the probability for STIs positivity increased.

Among the various cut-offs for the number of sex partners, the multivariable analysis demonstrated that both 1 and 5 sexual partners represented crucial cut-offs; women with >1 partner had almost 11 times elevated odds of harboring an STI (OR: 11.22, 95%CI: 3.17–39.76, *p* = 0.0002) and for >5 partners the odds were almost three times higher compared with those with ≤5 lifetime partners (OR: 2.97, 95%CI: 1.36–6.48, *p* = 0.0062). Furthermore, a partner change during the last year before the STI assessment resulted in more than 3 times higher odds for STI positivity (OR: 3.17, 95% CI: 1.57–6.40, *p* = 0.0013).

Neither smoking status, abnormal colposcopic impression, nor a positive HPV DNA or APTIMA test emerged as an important STI predictor; however, non-HPV vaccination status (OR: 2.51, 95% CI: 1.41–4.49, *p* = 0.0019) was correlated with a positive STI outcome. The overall concordance for STI status prediction of this logistic regression model was 74.1%, and the AUC or the ROC curve was 74.4%, indicative of a very good predictive ability.

Characteristic plots depicting graphically the impact of parity, number of sex partners, partner change during the last year, vaccination, and an abnormal test Pap are illustrated in Figure 2.

#### 2.2.3. Performance of HPV DNA and APTIMA Assays for the Detection of STI Positivity

In this cohort, from the 184 cases with negative HPV DNA tests, 120 also tested negative for STIs (True Negative—TN), whereas from the 152 women with positive HPV DNA tests, 72 were also positive for STIs (True Positive—TP). Additionally, of the 244 APTIMA-negative cases, 154 were also negative for STIs (TN), whereas from the 92 HPV APTIMA-positive individuals, 46 were also positive for STIs (TP). These results, along with the False Positive cases (FP, which refer to cases tested positive for HPV DNA or APTIMA tested negative for STIs) and the False Negative cases (FN, which refer to cases tested negative for HPV DNA or APTIMA tested positive for STIs, are summarized in Table 5 and Table 6.

## 3. Materials and Methods

### 3.1. Study Population—Inclusion and Exclusion Criteria

This work represents a post hoc analysis of supplementary data collected during an older prospective study. This study cohort’s characteristics have been described in a previous publication but are summarized herein for clarity [9]. In brief, we embarked on a prospective pragmatic observational study within the framework of a multidisciplinary research protocol in cervical pathology (Ministry of Education and Religious Affairs) under the frame of the HPVGuard study project (http://HPVGuard.org, accessed on 20 August 2024, Project Number: 11ΣΥΝ_10_250, Cooperation framework, Protocol Number: E_DE—ETAK 1788/1-10-2012). We enrolled asymptomatic women of reproductive age referred for colposcopy in a health setting affiliated with the University Hospital of Larisa, covering large areas of mainland Central Greece. Most patients were referred because of abnormal cytology. A single board-accredited experienced colposcopist performed all related procedures. We included all women of reproductive age harboring any grade of cytological abnormalities and/or abnormal colposcopy (LSIL/HSIL). We also included individuals testing positive for HPV DNA as measured by the CLART-2 HPV test^®^ (Genomica, Madrid, Spain) as well as those having a valid (positive or negative) mRNA E6 and E7 test as tested by the APTIMA ^®^ HPV Assay (Hologic, Marlborough, MA, USA); all samples were obtained at the initial visit. No eligibility restrictions regarding the anti-HPV vaccination status applied; however, individuals who were pregnant at the time of enrolment, those who had previously undergone conservative surgical cervical treatments, and those who had been previously reviewed in colposcopy for abnormal cytology were excluded. All participants signed the informed consent form, whereas procedures complied with the Helsinki Declaration.

### 3.2. Study Protocol

Based on this study’s protocol, a detailed medical and gynecological history was obtained at the first visit in all women, covering demographic aspects such as age at coitarche, parity, number of lifetime sexual partners, recent changes in sexual partnership, use of condoms, smoking status and HPV immunization.

In all participating individuals, a liquid-based cytology (LBC) sample was obtained using a Rovers™ Cervix brush just prior to the colposcopic evaluation. This was transferred in a single PreservCyt solution and subsequently underwent cytological and bio-molecular analysis with validated HPV assays, as follows:HPV DNA genotyping (CLART-2 HPV test^®^ (Genomica, Madrid, Spain)).Detection of E6/E7 mRNA from the 14 high-risk HPV types (APTIMA^®^ HPV Assay, (Hologic, Marlborough, MA, USA)).Molecular Bacterial STI detection utilizing QIAamp DNA Mini kit, focusing on *Chlamydia trachomatis* (Ct), *Mycoplasma genitalium* (Mg), *Mycoplasma hominis* (Mh), or *Ureaplasma* spp. (QIAGEN N.V., Hilden, Germany).

These molecular platforms represented standard assays utilized in the contributing laboratories. The cytological examination was reported according to the Bethesda classification (TBS 2014 system) [13]. HPV DNA extended genotyping using the CLART^®^ HPV2 platform (recently commercially discontinued) detects simultaneously 35 different HPV genotypes (of high, intermediate, and low risk) by PCR amplification of a fragment within the highly conserved L1 region of the virus. The APTIMA^®^ HPV Assay allows the identification of E6 and E7 mRNA of the 14 high-risk HPV types (16, 18, 31, 33, 35, 39, 45, 51, 52, 56, 58, 59, 66 and 68) utilizing a transcription-mediated amplification of viral mRNA; testing and analysis are performed on an automated system (Panther, Hologic). All specimens were centrally analyzed for cytological and biomolecular evaluation at the Athens University General Hospital “Attikon.” Bacterial STI detection utilizing the QIAamp DNA Mini kit was concurrently performed in a certified private sector molecular laboratory undergoing regular audits; this bacterial STI dataset was only subsequently evaluated for secondary, post hoc analysis presented in this paper.

Women with abnormal referral cytology, as well as those who tested HPV DNA and/or APTIMA positive, underwent colposcopic evaluation. Women with normal colposcopy had no biopsies obtained and were referred for a repeat cytological and colposcopic assessment six months later. Some individuals with low-grade colposcopic impressions underwent single or multiple-punch biopsies based on the overall clinician’s judgment. For colposcopic impressions suggestive of high-grade disease, either multiple biopsies were obtained, or a loop excision of the transformation zone (LLETZ) of the cervix was carried out. The subgroup of women with low-grade colposcopic impression in which biopsies were omitted was counseled for review in the colposcopy clinic in six months-time with repeat cytology. In all HPV unvaccinated individuals, a strong recommendation for HPV vaccination was advocated, irrespective of the individual cytological, molecular, or colposcopic findings. This study’s protocol has been approved by the Greek Central Government and subsequently received additional approval from the coordinating authority “Attikon” University Hospital Ethics Committee (Code: EBD 623/14-5-13).

### 3.3. Statistical Analysis

Statistical analysis was performed utilizing the SAS for Windows 9.4 software platform (SAS Institute Inc., Cary, NC, USA). Descriptive values were expressed as mean ± standard deviation (SD), also reporting minimum and maximum values, or median and first and third quartiles (Q1, Q3) or, for completeness reasons, both. Frequencies and the relevant percentages are reported for categorical data. For the qualitative parameters, comparisons between groups were based on the chi-square test and, when required, the Fisher exact. For arithmetic data (such as parity or age), the Mann–Whitney U test was used since normality (as evaluated by the Shapiro–Wilk test) was not assured. The performances of demographic characteristics such as age, parity (term pregnancies), sexual behavior, coitarche, vaccination as well as medical record details: cytology, colposcopy, HPV DNA, and APTIMA results were set as primary study’s outcomes, mainly for their relation to bacterial STI positivity. To investigate possible confounders, we applied multivariable analysis using the logistic regression approach. All tests were two-sided, and the significance level was set to α = 0.05.

### 3.4. Outcomes

Our study focused on investigating the co-variation of STI positivity and negativity with the results of the preselected HPV molecular assays (HPV DNA and APTIMA) in terms of their performance characteristics (positive and negative predictive values PPV and NPV correspondingly).

## 4. Discussion

This is probably the first study assessing the impact of multiple bacterial STI expression on cervical APTIMA HPV positivity rates in a colposcopic population. To the best of our knowledge, only one longitudinal study investigating APTIMA results simultaneously with bacterial STIs (focusing on Ct, Ng, Tv, and Mg) has been previously conducted; this has utilized the integrated APTIMA platform in samples obtained from Kenyan female sex workers, albeit without factoring Uu expression [14]. These authors concluded that recent or concurrent Ct infection was associated with more prolonged hrHPV expression.

In this study, we aimed to compare the performance of validated HPV mRNA and HPV DNA assays in terms of their “predictive ability” to identify concurrent STIs. Obviously, neither of these HPV assays represents an STI detection test; however, their results might contribute to overall clinical judgment or as input in a computerized algorithm mandating a more thorough diagnostic workup in selected individuals. The APTIMA HPV assay, which detects E6/E7 mRNA transcripts associated with the presence of persistent HPV infections, has been shown in a number of studies and meta-analyses to achieve higher specificity and comparable sensitivity to most HPV DNA-validated assays for the detection of CIN2+ in both colposcopy referral and primary screening settings [5,15]. Recent research showcases APTIMA’s excellent longitudinal performance in protecting against severe cervical precancer, in addition to corroborating its very high NPV [16]. The APTIMA HPV test can augment the selection of patients requiring cervical biopsy, sparing histologic confirmation for those with minimal risks [16,17]. Notably, the APTIMA HPV assay usually displays lower positivity across LSIL histology, likely due to the transient nature of HPV infections, which do not express E6 and E7 mRNA [15].

In brief, in our material, the HPV DNA test illustrated better performance for the detection of STI positivity than the APTIMA test (sensitivity 52.94% vs. 33.82%), whereas conversely, the APTIMA test had better performance in the detection of STI-negative cases than the hrHPV DNA test (specificity 77% vs. 60%); however, the two methods did not differ in terms of PPV, NPV, and overall accuracy. Given this study’s characteristics that are conducted in a referral population (colopscopy patients) and not a screening population, this could represent an anticipated finding. By identifying active, integrated infections that are more likely to cause cellular changes, the APTIMA test is more closely associated with the risk of cervical (pre)cancer. Obviously, if more hrHPV infections are detected, a stronger association with other STIs might emerge [5,15,18]. It is also likely that the relatively few total HSIL cases of this cohort affected overall APTIMA positivity rates [18].

In the recent study conducted in similar settings by Martinelli et al., in spite of the relatively high prevalence of hrHPV and STIs co-infections (predominantly with Up), no statistically significant association was documented between co-infections and corresponding abnormal colposcopy findings [19]. Furthermore, in a comprehensive SLR and meta-analysis by Ye et al., Uu imposed a significantly increased overall risk of HPV infections compared with Uu-negative women, with a substantially increased risk, particularly for hrHPV infections. In addition, Uu infection was associated with a significantly increased risk of abnormal cervical cytopathology, both LSIL (OR 2.02) and HSIL (OR 1.91) [20]. These authors suggest a possible mechanism of the association between Uu infection and abnormal cervical cytopathology related to the combination of several complex infection-associated inflammatory responses involving the production of reactive oxidative metabolites, increased expression of cytokines, chemokines, and growth and angiogenic factors, decreased cell-mediated immunity, and the generation of free radicals [20]. Perhaps the initial steps of HPV-mediated carcinogenesis are helped by a state of cervical inflammation, driven predominantly by the hormonal milieu, regulatory cytokines, and chemokines, as well as multiple cervicovaginal microorganisms [1]. The presence of Uu may play a role both in initiating cellular anomalies and in viral persistence [21,22,23,24,25,26,27]. An attractive hypothesis is the possibility that a gene or group of interrelated genes might be present in pathogenic ureaplasmas and absent in commensal ureaplasmas, which have not yet been distinguished by examination at the species or serovar level [28].

We have previously studied this cohort without focusing on STI detection, with findings corroborating previous epidemiologic research [9]. In this study, a partner change during the last year prior to HPV DNA assessment contributed to almost 2.5 times higher odds for DNA positivity (*p* = 0.0006); at that time, we remarked on the necessity for the development and validation of cervical scoring systems quantifying lifestyle factors impacting cervical precancer risk. These findings parallel previously published literature: notably, the older large epidemiological Finish study by Kataja et al. identified the number of sexual partners during the past 2 years as the strongest independent risk factor for genital HPV infections (adjusted odds ratio = 12.1) [29]. In a recent study conducted in Korea, the number of sexual partners in the previous 2 years also emerged as a determining factor for hrHPVs; because of the high rates of asymptomatic infection with HDC-Uu in this cohort, the authors consider that these bacteria should be eradicated whatsoever for the prevention of persistent HPV infection and subsequent CIN [26].

Recently, we have published our findings on possible associations of prevalent STIs with cervical cytology aberrations and extended HPV DNA genotyping results in the larger cohort studied in Greece so far [1]. In this high STI prevalence population (especially *Ureaplasma* spp.), a higher mean age was associated with negative STI testing. Interestingly, bacterial STI positivity illustrated significant heterogeneity between cases with NILM and LSIL cytology, with 28.88% of NILM and 46.33% of LSIL cases harboring an STI, respectively (*p* < 0.05).

In the virtues of this research it is among the first published studies assessing the co-variation of bacterial STI expression with APTIMA positivity. All cytological and molecular assessments (both for bacterial STIs as well as HPV) have been performed utilizing validated molecular assays in accredited laboratories undergoing regular external QA. The robust extended HPV genotyping assay utilized enhanced the emergence of subtle alterations attributable to low-risk HPV genotypes. Finally, the vaccination status of the enrolled patients has also been carefully documented.

Several non-anticipated factors resulted in significant study limitations. Notably, our results are almost certainly hampered by the few high-grade histology cases, this probably representing the main study’s drawback. It is extremely likely these outcomes would be diversified in another cohort enriched in HSIL disease, as is often documented in colposcopic populations. Importantly, we did not account for vaginal microecology, which is a most evolving field. We also did not assess syphilis or Ng prevalence; however, their anticipated impact would be marginal because of their low prevalence, also having been previously described regionally by other authors [19,30,31]. Finally, a possible limitation is that our results might be affected by the disproportionally low Ct and high Ureaplasma prevalence in our material, an epidemiological trend that has been corroborated in larger national studies [1].

Variations in STI prevalence and management strategies among different countries are attributable to differences in surveillance systems performance, STI testing policies, access to testing, as well as the availability of sensitive diagnostic techniques and algorithms [32,33,34]; however, in any setting, it is paramount to avoid detrimental surgical excisional overtreatments in cases of transient cervical HPV infections prone to regress. The challenges for STI management observed during epidemics such as COVID-19 should serve as lessons informing the development of effective interventions to mitigate their impact and enhance future public health preparedness [12,32,35]. In settings with high HIV prevalence, as well as in LMIC, there is an urgent need for STI screening together with an intensive scale-up of primary and secondary cervical cancer prevention programs [36]. Lastly, the potential for hrHPV and STI co-detection in vaginal or urine self-sampling material could eventually represent a ground-breaking strategy [1,33,34,37,38].

## 5. Conclusions

In this colposcopy patient population, the hrHPV DNA test illustrated better performance for the prediction of STI positivity than the corresponding APTIMA test (sensitivity 52.94% vs. 33.82%), while inversely APTIMA test was more indicative of bacterial STI negativity than the hrHPV DNA test (specificity 77% vs. 60%). In addition, no significant differences between these two molecular assays were documented in terms of PPV, NPV, and overall accuracy. Despite our clinical hypothesis that the APTIMA HPV assay might exhibit an enhanced predictive value for bacterial STI co-infections in this reproductive age cohort was not affirmed, this might be corroborated in other populations of referral cohorts with different demographic characteristics, hrHPV genotype distribution and overall background risk.

The high prevalence of STIs commonly found in hrHPV-positive women, as well as those referred for colposcopy who are, per definition, at elevated risk of developing cervical disease, indicate that comprehensive screening for genital infections in these populations may be justified [19,30,37]. With ongoing research in anti-STI vaccines currently under development, a possible co-administration with the HPV vaccine might emerge as a cost-effective public health intervention [1,37,39,40].

## Figures and Tables

**Figure 1 ijms-25-13146-f001:**
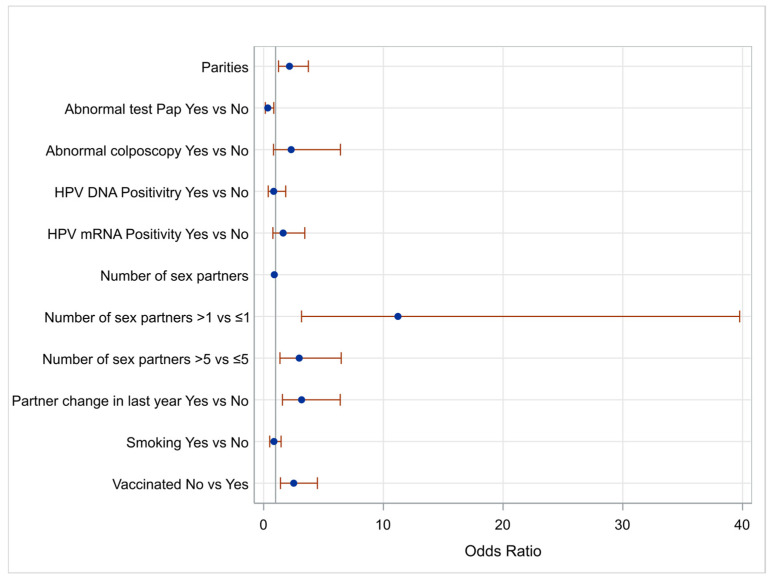
Odds Ratios and 95% confidence limits of parameters affecting STI positivity at univariate analysis.

**Figure 2 ijms-25-13146-f002:**
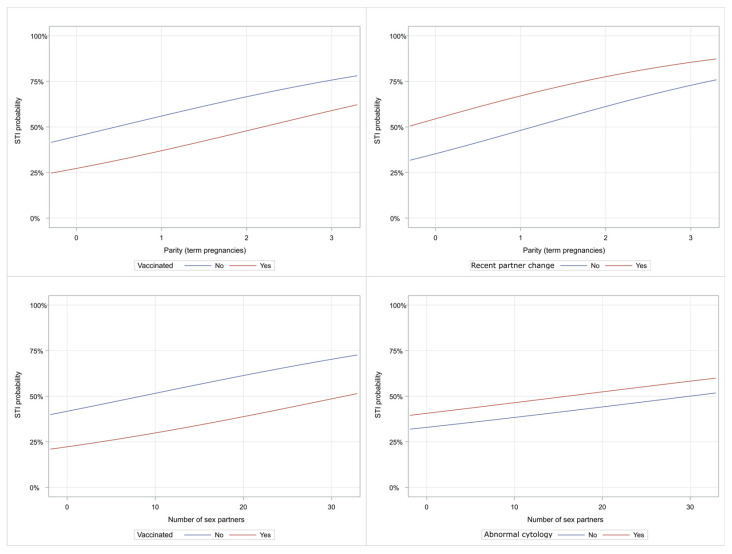
Characteristic curves illustrate the probability of STI positivity. (**Upper left**): parity and vaccination effect on STI positivity; (**Upper right**): parity and recent partner change effect on STI positivity; (**lower left**): number of sex partners and vaccination status on STI positivity and (**lower right**): number of sex partners and abnormal cytology role in STI positivity risk. The Vertical axis shows the probability of STI positivity.

**Table 1 ijms-25-13146-t001:** Demographic characteristics of the study population.

Characteristic	
Number of women (N)	336
Age (mean ± SD, minimum, maximum)	28.8 ± 6.3, 18–48
Smoking (N, %)	124 (36.9%)
Parity (term pregnancies) (mean ± SD, minimum, maximum)	0.16 ± 0.57, 0–3
HPV vaccination (N, %)	116 (34.52%)
Cervarix (HPV2)	22 (18.97%)
Gardasil 4 (HPV4)	94 (81.03%)
Number of lifetime sexual partners (mean ± SD, minimum, maximum)	6.0 ± 4.9, 1–30
Condom use percentage (mean ± SD, minimum, maximum)	29.8% ± 32.1%, 0–100%
0–19%	150 (44.64%)
20–39%	34 (10.12%)
40–59%	54 (16.07%)
60–79%	48 (14.29%)
80–99%	36 (10.71%)
100%	14 (4.17%)
Change in sexual partner within one year before inclusion in this study (N, %)	56 (16.7%)

**Table 2 ijms-25-13146-t002:** Cervical status and corresponding medical history of this study’s population.

LBC cytology results	(N, %)
NILM	156 (46.4%)
ASCUS	40 (11.9%)
LSIL (including HPV)	132 (39.3%)
ASC-H	none
HSIL	8 (2.4%)
HPV DNA result	(N, %)
Negative	184 (54.8%)
Positive	152 (45.2%)
High risk (from the 152 (+)ve samples)	146 (96.1%)
HPV mRNA result	(N, %)
Negative	244 (72.6%)
Positive	92 (27.4%)
Colposcopic Impression	(N, %)
Normal colposcopic findings and Mature Metaplasia	126 (37.5%)
Suggestive of LSIL (incl. HPV effect)	204 (60.7%)
Suggestive of HSIL	6 (1.8%)
Overall STIs positivity	(N, %)
Total STI-positive individuals	136 (40.5%)
Ct	2 (0.6%)
Mh	12 (3.6%)
*Ureaplasma* spp.	102 (30.4%)
Mg and *Ureaplasma* spp.	4 (1.2%)
Mh, *Ureaplasma* spp.	16 (4.8%)
Negative	200 (59.5%)
Potential coverage by the nonavalent vaccine (N, %) for women testing HPV DNA (+)ve	(N, %)
Full	56 (36.8%)
Partial	36 (23.7%)
No	60 (39.5%)
Histology results	(N, %)
Unavailable * (No biopsy specimen obtained)	116 (34.5%)
Negative/No documented SIL (Chronic cervicitis, etc)	42 912.5%)
LSIL (includes HPV)	170 (50.6%)
HSIL	8 (2.4%)
SCC	0

* Histological examination was not performed; these cases presented with normal cytology and colposcopy and tested HPV DNA negative.

**Table 3 ijms-25-13146-t003:** Statistical comparisons of STI positive vs. negative results with the other recorded variables.

	STIs Positivity vs. Parameter Levels *			
	Negative (Ν = 200)	Positive (Ν = 136)	*p* Value	OR (95% CI)
Parity (term pregnancies) (mean ± SD)	0.09 ± 0.40	0.26 ± 0.74	0.0069	NA
LBC ^1^	NILM (100/50%)	NILM (56/41.2%)	0.0152 ^2^	NA
	ASCUS (20/10%)	ASCUS (20/14.7%)		
	LSIL (72/36%)	LSIL (60/44.1%)		
	HSIL (8/4%)	HSIL (0/0%)		
Colposcopic Severity Findings (LSIL including HPV effect/metaplasia)	NILM (86/43%),	NILM (40/29.4%)	0.0019 ^2^	NA
	LSIL (108/54%)	LSIL (96/70.6%)		
	HSIL (6/3%)	HSIL (0/0%)		
Abnormal colposcopic impression	114/57%	96/70.6%	0.0116	1.8 (1.1–2.9)
HPV DNA positivity	80/40%	72/52.9%	0.0193	1.7 (1.1–2.6)
High-risk HPV DNA (within the HPV-positive group)	76/95%	70/97.2%	0.4823	1.8 (0.3–10.4)
Potential coverage by the nonavalent vaccine (within the HPV-positive group) (covered, partially covered, non-covered)	30/37.5%, 14/17.5%, 36/45.0%	26/36.1%, 22/30.6%, 24/33.3%	0.1318	NA
Multiple HPV subtypes	30/37.5%	28/38.9%	0.8603	1.1 (0.6–2)
HPV mRNA positivity	46/23%	46/33.8%	0.0290	1.7 (1.1–2.8)
Number of lifetime partners (Median, Q1–Q3)	4.5 (2.5–7)	5 (3–9)	0.0246	NA
Lifetime Partners > 1	168/84%	132/97.1%	0.0001	6.3 (2.2–18.2)
Lifetime Partners > 2	150/75%	124/91.2%	0.0002	3.4 (1.8–6.8)
Lifetime Partners > 5	62/31%	60/44.1%	0.0141	1.8 (1.1–2.8)
Partner change during last year	24/12%	32/23.5%	0.0054	2.3 (1.3–4)
Condom use percentage (Median, Q1–Q3)	20 (0–60)	40 (0–60)	0.3208	NA
Condom use	84/42%	62/45.6%	0.5149	1.2 (0.7–1.8)
Smoking	80/40%	44/32.4%	0.1539	0.7 (0.5–1.1)
HPV Vaccinated	84/42%	32/23.9%	0.0005	0.4 (0.3–0.7)
Genital Warts	12/6%	6/4.4%	0.5257	0.7 (0.3–2)

^1^ LBC: Liquid Based Cytology. ^2^ Fisher exact test, * Values reported: Number of cases (frequency) and relevant percentage.

**Table 4 ijms-25-13146-t004:** Multivariable analysis outcomes for STI positivity.

Effect	OR (95% CI)	*p* Value
Parity (term pregnancies)	2.16 (1.25–3.73)	0.0060
Abnormal test Papanicolaou	0.35 (0.14–0.85)	0.0202
Abnormal colposcopy	2.3 (0.82–6.41)	0.1121
HPV DNA positive	0.84 (0.38–1.84)	0.6615
APTIMA positive	1.63 (0.77–3.44)	0.2008
Number of sex partners	0.89 (0.82–0.97)	0.0073
Number of sex partners > 1	11.22 (3.17–39.76)	0.0002
Number of sex partners > 5	2.97 (1.36–6.48)	0.0062
Partner change in last year	3.17 (1.57–6.4)	0.0013
Smoking	0.86 (0.51–1.46)	0.5733
Non vaccinated	2.51 (1.41–4.49)	0.0019

**Table 5 ijms-25-13146-t005:** Expression of HPV DNA, APTIMA for STI-positive cases among the study population.

	TP	TN	FP	FN	Total
HPV DNA	72	120	80	64	336
APTIMA	46	154	46	90	336

TP: True positive; TN: True Negative; FP: False positive; FN: False Negative.

**Table 6 ijms-25-13146-t006:** Outline of the statistical comparisons of the performance indicators for the detection of STI positivity.

	HPV DNA	APTIMA	*p*-Value
Sensitivity	52.94%	33.82%	<0.00001
Specificity	60.00%	77.00%	<0.00001
Positive Predictive Value	47.37%	50.00%	0.4965
Negative Predictive Value	65.22%	63.11%	0.5687
False Positive Rate	40.00%	23.00%	<0.00001
False Negative Rate	47.06%	66.18%	<0.00001
Overall Accuracy	57.14%	59.52%	0.5287

## Data Availability

Data are available from the corresponding author upon a reasonable request.

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
