# Peer review of "Could APTIMA mRNA Assay Contribute to Predicting Cervical Bacterial Sexually Transmitted Co-Infections? A Colposcopy Population Studyâ€"

_ijms, 2024, doi:10.3390/ijms252313146_

Round 1
Reviewer 1 Report
Comments and Suggestions for Authors
I congratulate the authors on their work. The paper addresses the very important issue of co-infection. The influence of other infections on chronic HPV infection. Nowadays, many people know that persistent HPV infection is one of the most important risk factors, but there are also other very important factors.
I have a few comments regarding the work:
I would avoid the word pilot study in the title of the work. The group is not small.
I would avoid the word dysplasia; I suggest neoplasia.
The authors should use the LATS terminology in combination with the CIN terminology, which is very important and transparent: LSIL – CIN 1, HSIL – CIN 2, CIN 3?
In the introduction: I propose to write a few sentences about precancerous conditions LSIL - CIN 1 and HSIL CIN 2, CIN 3. Which changes may disappear spontaneously?
You can also write about genotyping in low- and high-grade lesions.
See publications:
Human papilloma virus genotyping in women with CIN 1 - Ginekol Pol 2010;81(9).
It would be interesting to analyze the regression of SIL changes in the case of co-infection with other HPV types.
Assessment of frequency of regression and progression of mild cervical neoplasia – LGSIL in women with positive high-risk HPV DNA test result – 2012.
Line 97-100 This sentence is incomprehensible - There is no information whether the patients were finally vaccinated or not. Vaccination affects the recurrence of the disease and persistent infections. In the tables that follow there is information about vaccination. I understand that this means that patients who got pregnant did not continue with HPV vaccinations.
In the future, we should limit smoking and e-cigarettes.
Table 2 The order of cytological diagnoses should be NILM, ASC-US, LSIL, ASC-H, HSIL.
Were there any situations where the molecular test result was non-diagnostic due to lack of material?
Were these cases excluded?
Were molecular tests performed from a single collection?
Very professionally prepared tables and figures.
Line 348 Should be HSIL.
The paper does not address the main disadvantage of the Aptima test - the test does not allow for the genotyping of HPV types. It has many numerous certificates but is not perfect. What we know is that simultaneous use of HPV DNA and mRNA tests does not confirm the type of infection or whether it is incidental and chronic.
Could the Covid-19 pandemic have affected the survey results? see publication: Women’s Healthcare Services since the COVID-19 Pandemic Outbreak in Poland. Int. J. Environ. Res. Public Health 2022, 19, 180. During the pandemic, many new recommendations have been created, including molecular diagnostics and self-sampling. You can cited in line 405.
I think that the conclusions are too long and there are too many citations in the conclusions themselves. The part about creating new diagnostic tests combined - detecting HPV infections and other SDIs should be in the discussion.
Please use uniform terminology and not HG once and then HSIL.
I recommend publishing it after making corrections.
Author Response
I congratulate the authors on their work. The paper addresses the very important issue of co-infection. The influence of other infections on chronic HPV infection. Nowadays, many people know that persistent HPV infection is one of the most important risk factors, but there are also other very important factors. I have a few comments regarding the work:
Thank you for dedicating your time and providing valuable expertise.
I would avoid the word pilot study in the title of the work. The group is not small.
Thank you for your comment. We have indeed revised the Article’s title.
I would avoid the word dysplasia; I suggest neoplasia.
Thank you for your comment. We have used the terms precancer/SIL.
The authors should use the LATS terminology in combination with the CIN terminology, which is very important and transparent: LSIL – CIN 1, HSIL – CIN 2, CIN 3?
Thank you for your comment. We have changed the terms in most instances.
In the introduction: I propose to write a few sentences about precancerous conditions LSIL - CIN 1 and HSIL CIN 2, CIN 3. Which changes may disappear spontaneously? You can also write about genotyping in low- and high-grade lesions. See publications: Human papilloma virus genotyping in women with CIN 1 - Ginekol Pol 2010;81(9). It would be interesting to analyze the regression of SIL changes in the case of co-infection with other HPV types.Assessment of frequency of regression and progression of mild cervical neoplasia – LGSIL in women with positive high-risk HPV DNA test result – 2012.
Thank you for your comment. We have altered the text and provided additional references.
Line 97-100 This sentence is incomprehensible - There is no information whether the patients were finally vaccinated or not. Vaccination affects the recurrence of the disease and persistent infections. In the tables that follow there is information about vaccination. I understand that this means that patients who got pregnant did not continue with HPV vaccinations.
Thank you for your comment. Changes to the text have been made accordingly.
In the future, we should limit smoking and e-cigarettes.
We agree with your point, thank you!
Table 2 The order of cytological diagnoses should be NILM, ASC-US, LSIL, ASC-H, HSIL.
Thank you. Cytological diagnoses are now stratified based on severity in the revised version.
Were there any situations where the molecular test result was non-diagnostic due to lack of material? Were these cases excluded?
No such cases were identified in our dataset, therefore no exclusions applied.
Were molecular tests performed from a single collection?
Yes, all assays were performed from material from a single collection in one ThinPrep vial.
Very professionally prepared tables and figures.
Thank you!
Line 348 Should be HSIL.
Thank you for the correction.
The paper does not address the main disadvantage of the Aptima test - the test does not allow for the genotyping of HPV types. It has many numerous certificates but is not perfect. What we know is that simultaneous use of HPV DNA and mRNA tests does not confirm the type of infection or whether it is incidental and chronic.
We agree on these limitations of the APTIMA assay. Thank you for your comment.
Could the Covid-19 pandemic have affected the survey results? see publication: Women’s Healthcare Services since the COVID-19 Pandemic Outbreak in Poland. Int. J. Environ. Res. Public Health 2022, 19, 180. During the pandemic, many new recommendations have been created, including molecular diagnostics and self-sampling. You can cite in line 405.
The recruitment of this cohort was completed before the COVID pandemic. In lines 404-407 a short mention regarding the effect of the COVID pandemic on STI’s management is cited.
I think that the conclusions are too long and there are too many citations in the conclusions themselves. The part about creating new diagnostic tests combined - detecting HPV infections and other SDIs should be in the discussion.
We have amended the text accordingly.
Please use uniform terminology and not HG once and then HSIL.
Thank you for noticing, this is corrected for LGSIL and HGSIL
I recommend publishing it after making corrections.
Thank you for your positive feedback and instructive comments to improve this work.
Reviewer 2 Report
Comments and Suggestions for Authors
Dear authors,
The results of studies like this can be published. Some ideas and conclusions, in my opinion, should be rephrased. I recommend to reconsider the article after minor revision.
Introduction and further
The abbreviations are not explained: STI, HPV, DNA, mRNA, LGSIL, HGSIL
Introduction, line 62
Evolution in cervical prevention strategies has mandated a global shift towards HPV-based approaches, soon to affect younger individuals entering screening.
Note:
HPV-based screening, including self-sampling, is mainly aimed at older women who do not regularly participate in cervical cancer screening examinations. Change it or delete.
Introduction, line 62
….to assess subtle differences in HPV DNA and APTIMA positivity ….
Note: ….to assess subtle differences between hrHPV DNA and hrHPV mRNA positivity ….
Study protocol, line 134
Women with abnormal referral cytology of any grade as
Note: Do have cytology have any grades? Delete it.
Study protocol, line 141
The subgroup of women with low-grade colposcopy in which ….
Note: It's not the exact terminology, but what you used before is better: low-grade colposcopic impression
Results, line 173
… with the quadrivalent Gardasil (add vaccine)
Results, line 184
More than half of the individuals (53.6%) presented with aberrant cytology results, while 62.5% had abnormal colposcopic findings.
Note: add the real numbers (which they are shown in more detail in Table 2) into the text to clarify,
Discussion, line 331
In brief, in our material, the HPV DNA test illustrated better performance for the detection of STIs positivity than the APTIMA test (sensitivity 52.94% vs. 33.82%) whereas conversely the APTIMA test had better performance in the detection of STI negative cases than the HPV DNA test (specificity 77% vs 60%). However, the two methods did not differ in terms of PPV, NPV and overall accuracy. Phenomenally, this represents an unexpected result, given APTIMA’s enhanced predictive value compared with that of HPV DNA test, especially given the study’s characteristics - conducted in a referral population (colopscopy patients) and not a screening population [4,14,17]. Possibly, this might be attributed to the relatively few total HG cases of this cohort, affecting overall APTIMA positivity rates [17].
Note:
This is not a phenomenally unexpected result. HPV DNA test detects the presence of viral DNA (whether or not the virus is active), and is more useful for detecting the overall presence of high-risk HPV types. HR HPV mRNA test detects the presence of mRNA produced by the virus, which indicates active viral expression. The true is that mRNA test is more closely associated with the risk of cervical (pre)cancer because it identifies active infections that are more likely to cause cellular changes. On the other side, HPV DNA test is more commonly used for initial screening because it identifies HPV infections early, even before they cause significant cellular changes. But if we consider that the presence of HR HPV is associated with the presence of other STIs, it is logical that if we detect all HR HPV infections, the association with other STIs will be higher (similar risk factors). Also because HPV is the most common human STI. Rephrase the text in this sense.
Conclusions, line 408
Despite our hypothesis that the APTIMA HPV assay might exhibit enhanced predictive value for bacterial STI co-infections in this colposcopic population of reproductive age was not affirmed in this study, this might be corroborated in other populations of referral cohorts with different demographic characteristics, hrHPV genotype distribution and overall background risk. The high prevalence of STIs commonly found in hrHPV-positive women as well as those referred for colposcopy who are per definition at elevated risk of developing cervi-cal disease, indicate that comprehensive screening for genital infections in these popula-tions may be justified [18,29,35]. With ongoing research in anti-STI vaccines currently under development, a possible co-administration with the HPV vaccine might emerge as a cost-effective public health intervention [1,35]. Furthermore, the potential for hrHPV and STI co-detection in vaginal or urine self-sampling material could eventually represent a ground-breaking strategy [1,32,33,35,36].
Note: This part belongs to the discussion. In the conclusion section, you should summarize the published results in 3-4 sentences. Rephrase the text in this sense.
Author Response
Dear authors, the results of studies like this can be published. Some ideas and conclusions, in my opinion, should be rephrased. I recommend to reconsider the article after minor revision.
Thank you for your comments and feedback, we hope that the revised manuscript has addressed your concerns.
Introduction and further
The abbreviations are not explained: STI, HPV, DNA, mRNA, LGSIL, HGSIL
Indeed, all these abbreviations were systematically quoted at the time of the initial submission at the last page of the manuscript; however they were apparently truncated during article uploading. They are now re-introduced in this revised version, following the Conclusions section.
Introduction, line 62
Evolution in cervical prevention strategies has mandated a global shift towards HPV-based approaches, soon to affect younger individuals entering screening.
Note:
HPV-based screening, including self-sampling, is mainly aimed at older women who do not regularly participate in cervical cancer screening examinations. Change it or delete.
Thank you, we have amended the text accordingly.
Introduction, line 62
….to assess subtle differences in HPV DNA and APTIMA positivity ….
Note: ….to assess subtle differences between hrHPV DNA and hrHPV mRNA positivity ….
Thank you for noticing. We have incorporated the correction.
Study protocol, line 134
Women with abnormal referral cytology of any grade as
Note: Do have cytology have any grades? Delete it.
We have changed accordingly.
Study protocol, line 141
The subgroup of women with low-grade colposcopy in which ….
Note: It's not the exact terminology, but what you used before is better: low-grade colposcopic impression
Thank you, we have changed accordingly.
Results, line 173
… with the quadrivalent Gardasil (add vaccine)
Thank you, we have changed accordingly.
Results, line 184
More than half of the individuals (53.6%) presented with aberrant cytology results, while 62.5% had abnormal colposcopic findings.
Note: add the real numbers (which they are shown in more detail in Table 2) into the text to clarify,
Thank you, we have changed accordingly.
Discussion, line 331
In brief, in our material, the HPV DNA test illustrated better performance for the detection of STIs positivity than the APTIMA test (sensitivity 52.94% vs. 33.82%) whereas conversely the APTIMA test had better performance in the detection of STI negative cases than the HPV DNA test (specificity 77% vs 60%). However, the two methods did not differ in terms of PPV, NPV and overall accuracy. Phenomenally, this represents an unexpected result, given APTIMA’s enhanced predictive value compared with that of HPV DNA test, especially given the study’s characteristics - conducted in a referral population (colopscopy patients) and not a screening population [4,14,17]. Possibly, this might be attributed to the relatively few total HG cases of this cohort, affecting overall APTIMA positivity rates [17].
Note:
This is not a phenomenally unexpected result. HPV DNA test detects the presence of viral DNA (whether or not the virus is active), and is more useful for detecting the overall presence of high-risk HPV types. HR HPV mRNA test detects the presence of mRNA produced by the virus, which indicates active viral expression. The true is that mRNA test is more closely associated with the risk of cervical (pre)cancer because it identifies active infections that are more likely to cause cellular changes. On the other side, HPV DNA test is more commonly used for initial screening because it identifies HPV infections early, even before they cause significant cellular changes. But if we consider that the presence of HR HPV is associated with the presence of other STIs, it is logical that if we detect all HR HPV infections, the association with other STIs will be higher (similar risk factors). Also because HPV is the most common human STI. Rephrase the text in this sense.
Thank you for your insightful comments. We have altered the text accordingly, integrating several of your points.
Conclusions, line 408
Despite our hypothesis that the APTIMA HPV assay might exhibit enhanced predictive value for bacterial STI co-infections in this colposcopic population of reproductive age was not affirmed in this study, this might be corroborated in other populations of referral cohorts with different demographic characteristics, hrHPV genotype distribution and overall background risk. The high prevalence of STIs commonly found in hrHPV-positive women as well as those referred for colposcopy who are per definition at elevated risk of developing cervical disease, indicate that comprehensive screening for genital infections in these populations may be justified [18,29,35]. With ongoing research in anti-STI vaccines currently under development, a possible co-administration with the HPV vaccine might emerge as a cost-effective public health intervention [1,35]. Furthermore, the potential for hrHPV and STI co-detection in vaginal or urine self-sampling material could eventually represent a ground-breaking strategy [1,32,33,35,36].
Note: This part belongs to the discussion. In the conclusion section, you should summarize the published results in 3-4 sentences. Rephrase the text in this sense.
Thank you for your point. We have altered the text to ameliorate the correct structure of the manuscript.